# Pharmacological Inhibition of Myostatin in a Mouse Model of Typical Nemaline Myopathy Increases Muscle Size and Force

**DOI:** 10.3390/ijms242015124

**Published:** 2023-10-12

**Authors:** Johan Lindqvist, Henk Granzier

**Affiliations:** Department of Cellular and Molecular Medicine, University of Arizona, Tucson, AZ 85724, USA; johanlindqvist@email.arizona.edu

**Keywords:** nemaline myopathy, myostatin, nebulin

## Abstract

Nemaline myopathy is one of the most common non-dystrophic congenital myopathies. Individuals affected by this condition experience muscle weakness and muscle smallness, often requiring supportive measures like wheelchairs or respiratory support. A significant proportion of patients, approximately one-third, exhibit compound heterozygous nebulin mutations, which usually give rise to the typical form of the disease. Currently, there are no approved treatments available for nemaline myopathy. Our research explored the modulation of myostatin, a negative regulator of muscle mass, in combating the muscle smallness associated with the disease. To investigate the effect of myostatin inhibition, we employed a mouse model with compound heterozygous nebulin mutations that mimic the typical form of the disease. The mice were treated with mRK35, a myostatin antibody, through weekly intraperitoneal injections of 10 mg/kg mRK35, commencing at two weeks of age and continuing until the mice reached four months of age. The treatment resulted in an increase in body weight and an approximate 20% muscle weight gain across most skeletal muscles, without affecting the heart. The minimum Feret diameter of type IIA and IIB fibers exhibited an increase in compound heterozygous mice, while only type IIB fibers demonstrated an increase in wild-type mice. In vitro mechanical experiments conducted on intact extensor digitorum longus muscle revealed that mRK35 augmented the physiological cross-sectional area of muscle fibers and enhanced absolute tetanic force in both wild-type and compound heterozygous mice. Furthermore, mRK35 administration improved grip strength in treated mice. Collectively, these findings indicate that inhibiting myostatin can mitigate the muscle deficits in nebulin-based typical nemaline myopathy, potentially serving as a much-needed therapeutic option.

## 1. Introduction

Nemaline myopathies (NM) are a group of rare congenital disorders that primarily impact skeletal muscles, leading to varying degrees of muscle weakness among patients [1,2]. The most severely affected individuals experience reduced fetal movements and require ventilatory support due to a lack of spontaneous breathing at birth. However, most patients exhibit moderate weakness, substantially restricting their daily activities [3,4]. A recent study conducted on a Brazilian cohort revealed that approximately one-third of individuals with nebulin-based NM (NEM2) were unable to walk unassisted, and over half of them relied on ventilatory support [5]. Another study indicated that all NM patients exhibited respiratory muscle weakness [6]. In addition to skeletal muscle weakness, the classic diagnostic feature of NM is the presence of rod-shaped protein aggregates known as nemaline bodies. These aggregates typically originate at the Z-disc but can also be found in subsarcolemmal regions and nuclei [7]. NM is caused by mutations in genes responsible for encoding proteins associated with thin filaments, as well as their assembly, maintenance, and turnover. The most frequently mutated gene, occurring in approximately 35% of cases, is NEB, which encodes the nebulin protein [3,8].

Nebulin predominantly consists of repetitive modules that are approximately 30 amino acids long with each containing an actin binding site. Seven of these modules combine to form super-repeats over a sizable portion of nebulin’s length [9]. The number of super-repeats varies among species, with larger animals typically having more super-repeats [10]. A recent study utilizing cryo-electron tomography revealed that nebulin is situated within the grooves between the two actin strands. However, it does not directly interact with tropomyosin or myosin heads but instead appears to interact with a linker region in troponin T [11]. Nebulin plays a crucial role in facilitating cross-bridge formation and cycling, ensuring proper regulation of thin filament length, and contributing to Z-disc organization and alignment of myofibrils [9,12,13,14,15].

Currently, there is no specific treatment available for NM, and the existing interventions mainly focus on providing functional support, thus, there is an urgent need to develop therapies for NM. Myostatin, also referred to as GDF8, is a protein that negatively regulates muscle mass [16]. Animals and humans with a myostatin null-genotype exhibit “double-muscling” [17,18]. This discovery has spurred the development of compounds aimed at inhibiting the function of myostatin. Clinical trials have primarily focused on testing myostatin inhibitors as potential treatments for various neuromuscular disorders, including Duchenne muscular dystrophy (DMD), facioscapulohumeral muscular dystrophy, inclusion body myositis, limb-girdle muscular dystrophies, as well as conditions like sarcopenia and cachexia [16]. Inhibition of myostatin presents an appealing therapeutic approach for NM, as it has the potential to increase muscle mass in patients who typically have a very lean musculature, which contributes to overall muscle weakness. The additional muscle mass could potentially help alleviate the weakness, leading to improved performance in daily activities and an enhanced quality of life. 

Hence, we conducted experiments using a mouse model that mimics the typical form of NM. These mice carry compound heterozygous mutations in the nebulin gene. In our study, we treated these mice with mRK35, a mouse antibody specifically targeting myostatin [19,20]. The compound heterozygous (Compound-Het) mice exhibit a normal lifespan, but they are smaller in size and display reduced grip strength, muscle weakness, and muscle atrophy/hypotrophy. Additionally, these mice exhibit nemaline rods and structural abnormalities characterized by disorganized sarcomeres and Z-disc streaming [20]. To assess the effects of mRK35 on this mouse model, we evaluated muscle size and function, both in vitro and in vivo.

## 2. Results

### 2.1. Body Weights

We have housed Compound-Het mice until 14 months of life and observed no excess mortality compared to wild-type mice. All mice included in the study survived until euthanasia and tissue harvest. Throughout the study, body weights were recorded, and growth curves were generated using Gompertz’s function. Consistent with previous findings [20], Compound-Hets exhibited smaller body sizes compared to their wild-type littermates. However, at the conclusion of the study, treatment with mRK35 resulted in a 9% increase in body weights for wild-type mice (Figure 1A) and a 12% increase for Compound-Hets (Figure 1B). Remarkably, at the end of the mRK35 treatment, the body weight deficit observed in Compound-Hets compared to untreated wild-type mice had been eliminated (Figure 1C). 

### 2.2. Tissue Weights

To determine whether the observed greater body mass is accompanied by increased muscle mass, we examined a variety of striated muscle types. Muscle weights were measured and normalized to tibia lengths. Most muscles in Compound-Het mice were atrophic/hypotrophic compared to wild-type mice while tibialis cranialis and diaphragm muscles were hypertrophic (Appendix A), as reported before for this model [20]. In wild-type animals, mRK35 significantly increased muscle mass by on average 22%, except for soleus (Figure 2A, relative values shown). The mRK35 treatment of Compound-Hets resulted in ~19% greater weights of all examined compared to vehicle-treated Compound-Hets (Figure 2B, relative values shown; the dotted line indicates untreated wild-type mice). Notably, the treatment with mRK35 in Compound-Het mice either restored muscle weights to the muscle weights of untreated wild-type mice or exceeded that of untreated wild-type mice (Figure 2B). We did not find any effect of mRK35 on the heart chamber weights, in neither wild-types (Figure 3A) or Compound-Hets (Figure 3B). Thus, mRK35 specifically affects skeletal muscle.

### 2.3. Cross-Sectional Area

To examine the fiber type specificity of mRK35, we measured the minFeret diameter of individual muscle fibers. Immunofluorescence images of cryosections from EDL muscles stained with antibodies specific against the different fiber types are shown in Figure 4A. In wild-type animals, mRK35 treatment increased the diameter of type IIB fibers by 21.1% (Figure 4B), while in Compound-Hets the diameter of both type IIA and IIB were 21.6% and 16.9% larger, respectively, after mRK35 administration (Figure 4C). The increase in minFeret diameter matches the muscle weight increase suggesting that a large fraction of the mass increase of mRK35-treated muscles is likely due to hypertrophy of IIA and IIB fiber types. 

### 2.4. Myostatin Protein Levels and Myosin Heavy Chain Distribution

To evaluate whether pharmacological myostatin inhibition affected myostatin protein levels, we performed Western blotting on the EDL muscle lysates. Myostatin protein levels were greatly decreased in untreated Compound-Hets compared to untreated wild-types. In wild-type mice, mRK35 increased the myostatin protein levels fourfold. No effect of myostatin inhibition was seen in Compound-Hets (Appendix A). We also studied whether the myosin heavy chain distribution was affected by mRK35 in EDL muscles. At baseline, Compound-Het mice had a decreased proportion of type IIB myosin while the proportions of type IIA/X (we were not able to resolve their individual isoforms) and type I myosin were increased as described before [20]. We observed a small increase in the proportion of fast type IIB myosin with a concomitant decrease in type IIA/X myosin proportions in mRK35-treated wild-types. No effect of myostatin inhibition was seen on myosin heavy chain distribution in Compound-Hets (Appendix A).

### 2.5. Intact Muscle Mechanics

To investigate whether the muscle mass increase after mRK35 treatment was functional, we initially measured in situ plantarflexion force generation using the foot plate system described in Lindqvist et al. [20]. After mRK35 treatment, the hindlimbs of the mice were challenging to correctly position in the setup and the electrodes were difficult to place in proximity to the sciatic nerve due to the increased muscle size. We concluded that the results of these experiments were not reliable. Thus, we decided to perform intact muscle experiments on Extensor Digitorum Longus (EDL) muscles instead. Isolated muscles were lengthened to optimal length, as determined by twitch stimulations. The optimal length of EDL did not differ between genotypes at baseline and statistical testing using 2-way ANOVA revealed a significant treatment effect of mRK35 on optimal length (Figure 5A). However, multiple comparison testing with Tukey’s correction method across all groups showed only a trend (*p* = 0.06) toward longer optimal length in wild-type mice (Figure 5A). The physiological cross-sectional area, pCSA, of EDL muscle was smaller in Compound-Hets compared to wild-types (Figure 5B). Treatment with mRK35 increased pCSA by 23% and 21% in wild-type and Compound-Het mice, respectively. Notably, the pCSA of mRK35-treated Compound-Het mice were similar to untreated control mice (Figure 5B). The maximal isometric force produced during tetanus was 33% lower in Compound-Hets compared to controls at baseline (Figure 5C). The mRK35 treatment increased tetanic force by 14% and 20% in wild-type mice and Compound-Het mice respectively (Figure 5C). In line with our previous study [20], the specific force (force normalized to pCSA) in untreated mice was decreased by 22% in Compound-Hets compared to wild-types, and no effect of mRK35 was observed on the specific force in neither wild-type nor Compound-Het mice (Figure 5D). We also examined whether mRK35 had any effect on contraction kinetics. Time to half-maximal force normalized to the produced force was unchanged between wild-type and Compound-Hets (Appendix A), while time to half relaxation normalized to force was significantly slower in Compound-Hets compared to wild-type mice (Appendix A). mRK35 had no effect on the kinetics of contraction (Appendix A).

### 2.6. Running Wheel and Grip Strength Studies

To study whether mRK35 treatment had functional effects at the whole animal level, we performed a running wheel study. No differences in average running speed, running distance, or run time were observed between healthy wild-type mice and Compound-Het mice; mRK35 treatment had no effect on average speed, running distance, or run time in neither WT or Compound-Het mice (Appendix A). Grip strength measurements throughout the study, starting just after weaning were also performed. At baseline, Compound-Het mice were significantly weaker compared to wild-type mice when performing pulls using all four limbs (Figure 6C). In wild-type mice, mRK35 treatment only significantly increased grip strength at the final test at four months of life (Figure 6A). Interestingly, at 2 months and at every following time point, mRK35-treated Compound-Het mice were significantly stronger than untreated Compound-Hets (Figure 6B). Further, grip strength in mRK35-treated Compound-Het mice approached the grip strength of untreated wild-type mice (Figure 6C).

## 3. Discussion

Myostatin, a member of the TGF-β superfamily, acts as a negative regulator of muscle size [21,22]. Many diseases are characterized by muscle atrophy and inhibition of myostatin has emerged as an appealing therapeutic approach that alleviates overall muscle weakness and enhances the performance of daily activities, thus improving quality of life [16,23]. In the case of NM, a condition marked by muscle weakness and muscle smallness due to atrophy/hypotrophy, there are currently no therapies available [3,8]. This makes myostatin inhibition an intriguing candidate for therapeutic evaluation in NM. We found that myostatin inhibition results in an increase in muscle weights across many skeletal muscles, without affecting the heart, and that it increases force in both wild-type and compound heterozygous NM mice. These findings indicate that inhibiting myostatin can mitigate muscle deficits in nebulin-based typical nemaline myopathy, potentially serving as a much-needed therapeutic option.

In this study, we investigated the effects of the myostatin inhibitors mRK35 on a mouse model that mimics the typical form of NM, and in this model, mRK35 treatment resulted in increased body weights (Figure 1B) and ~20% greater muscle mass due to fiber hypertrophy of type IIA and IIB fibers (Figure 2B and Figure 4C). Intact whole-muscle experiments on EDL revealed that both pCSA and tetanic force were increased by ~20% (Figure 5B, C). The specific force was unaffected by treatment indicating that mRK35 had no effect on the quality of muscle in Compound-Hets (Figure 5D). Following this, mRK35-treated Compound-Het mice displayed greater grip strength from 2 months of life lasting until the end of the study (Figure 6B). Interestingly, voluntary exercise performance was unaffected by mRK35, suggesting that the increased muscle strength of the level achieved does not have an impact on voluntary exercise performance (Appendix A). This is consistent with the lack of an exercise deficit in the Compound-Het mice under baseline conditions when comparing them to wild-type mice (Appendix A). 

In wild-type mice primarily type IIB fibers responded to mRK35 by increasing the MinFeret diameter (Figure 4A). Considering that healthy soleus muscles contain very few type IIB fibers [20,24] this explains why in mRK35-treated wild-type mice the slow-twitch soleus muscle did not increase in weight while other predominantly fast-twitch muscles had ~20% increased weights (Figure 2A). Similar findings have been observed in other studies on wild-type mice [21,25]. The fiber type-specific response to mRK35 can be explained by the high expression level of ActRIIb mRNA in fast mouse muscles [26,27]. Circulating myostatin is cleaved into its active form that binds to ActRII- and ActRIIB-receptors, which triggers multiple signaling cascades that regulate the expression of genes important for muscle growth and protein turnover rates. This includes regulating the expression of MuRF1 and MAFbx by FoxO1/3a dependent and independent mechanisms. Myostatin has also been found to regulate the IGF-1/AKT/mTOR-pathway through miRNA-486. Thus, myostatin regulates muscle mass through multiple signaling pathways [16,26,28]. Surprisingly, soleus muscles from Compound-Het mice had 12% greater mass after mRK35 treatment (Figure 2B). Additionally, in Compound-Hets type 2A fibers in EDL responded to mRK35 (Figure 4B) indicating that that this fiber type can also respond to myostatin inhibition in nebulin-based NM (NEM2) mice. A common finding in NEM2 is a shift towards slower fiber types [4,20,24] and mRK35′s effect on the slow-twitch soleus muscles from Compound-Het mice suggests that mRK35 could be more potent in muscles with typical NM than in healthy slow-twitch muscles. In our study, myostatin inhibition increased the fraction of type IIB myosin in wild-type mice and had no effect on Compound-Het mice (Appendix A). Constitutive loss of myostatin has been found to change myofiber composition in skeletal muscles by increasing the proportion of type IIB fibers, while concomitantly decreasing the fraction of slow type I fibers. This has been attributed to myostatin signaling during fetal development [29,30]. The limited effect on myosin isoform distribution observed in this study is likely linked to that the myostatin inhibition was started at 14 days of life, which is outside the postnatal period. Further studies delineating the molecular cause of these observations are warranted.

Three other studies have tested myostatin inhibition in mouse models of actin-based NM and NEM2 with diverse results [31,32,33]. In a previous study, conditional nebulin knock-out (NEB cKO) mice were treated with the myostatin inhibitor ActRIIB-mFc (RAP-031) from Acceleron Pharma. In that study, no beneficial effects of treatment were seen on body weight, grip strength, and myofiber size [31]. This lack of a response contrasts that of the present study and it is possible that the neutral effect in the previous study results from the much more severe and complex phenotype of the NEB cKO model [24,34,35]. Similarly, a divergent response to myostatin inhibition on muscle function was shown in two mouse models of actin-based NM with different disease severity [36,37]. mRK35 improved grip strength, increased muscle mass, and an absolute force of membrane-permeabilized single fibers while maintaining the specific force in the less severe Tg*ACTA1*^D286G^ mouse model [32]. Twice weekly injections of ActRIIB-mFC in Tg*ACTA1*^H40Y^ mice with severe NM resulted in larger muscles and improved survival of transgenic mice but had no effect on muscle force with a reduction in specific force suggesting worsened quality of contraction. Thus, it is possible that the difference in disease severity dictates the therapeutic effect of myostatin inhibition with a greater benefit in less severe forms of the disease. It cannot be excluded, although we consider it less likely, that the results of these studies depend on the form of myostatin inhibition. RAP-031 is an ActRIIB receptor decoy and mRK35 is a monoclonal antibody against myostatin. In summary, in less severe forms of NM, myostatin inhibition is effective in increasing muscle size and force. 

mRK35 was developed by Pfizer Inc. and exists in a humanized form called Domagrozumab [19]. It was evaluated in a phase 2 clinical trial for DMD, and although it failed to improve the primary endpoint (4-stair climb test), the secondary endpoints thigh muscle volume and muscle volume index were increased [38,39]. The pathological mechanisms in DMD and other dystrophies are significantly different from NM, which could influence the therapeutic efficacy of myostatin inhibition. DMD is a progressive disease characterized by fibrosis, fatty replacement, and loss of muscle tissue [40,41]. Dystrophin is a sarcolemma protein that is important for cell membrane stability and loss of it leads to sarcolemma tears, increased cytosolic calcium levels, and oxidative stress that results in muscle injury, regeneration, and the eventual death of the muscle cells. This triggers muscle wasting, fibrosis, and fat replacement seen in DMD [41]. It is possible that the increased muscle size, due to inhibition of myostatin, would put additional stress on the sarcolemma suggesting that DMD is possibly not an ideal target for a myostatin inhibition-based treatment. NM, on the other hand, is a thin filament disease with more than half of the patients having mutations in skeletal muscle α-actin or nebulin [3]. Many studies by us and others have shown that muscle weakness in NM originates at the myofilament level and that these mutations disrupt the finely tuned cross-bridge cycling mechanism that is responsible for generating muscle force [20,42,43,44,45,46]. Misalignment and structural abnormalities of the sarcomeres likely also contribute to muscle weakness, as do muscle hypotrophy/atrophy [20,24,46]. In contrast to dystrophies, the sarcolemma remains intact, and NM is non- or only slowly progressive for limb motor function [5]. Further, one study has found decreased levels of circulating myostatin and decreased myostatin mRNA in muscle biopsies of DMD patients that possibly explain the low clinical efficacy of myostatin inhibition in DMD [47]. Similarly, we found greatly decreased myostatin levels in Compound-Het mice compared to healthy mice, which could be related to the relatively mild disease in this model (Appendix A). Other studies have found increased myostatin protein levels in two mouse models of severe NM [31,33]. No change in myostatin was observed in a mouse model of mild actin-based NM. Additionally, myostatin positively regulates the expression of MuRF1 and MAFbx [16,26] and last year, we found that MuRF1 protein levels are elevated in a limited number of patients and two animal models of NEM2 [48]. Collectively, current data from animal experiments and clinical trials indicate that myostatin inhibition increases muscle size in animals and humans, but improvements in muscle function are disease-dependent [49]. These differences in pathological mechanisms between dystrophies and NM could offer a potential for myostatin inhibition as a therapy to enhance overall muscle force as the muscle size increases and thus have beneficial effects in NM. 

In summary, this study showed that weekly administration of mRK35, a myostatin antibody, can trigger increased muscle size and force in a mouse model that mimics both genetically and pathologically the most common form of NM. Grip strength, a common functional muscle test, was also improved with mRK35 treatment in Compound-Het mice. Taken together, our results indicate that pharmacological inhibition of myostatin signaling can be a therapeutic option in non-severe nebulin-based NM and further evaluation of mRK35 should be considered.

## 4. Materials and Methods

### 4.1. Animals

The mouse model for NEB compound heterozygosity has a point mutation in one of the NEB alleles corresponding to nebulin S6366I-substitution found in humans [50,51], while the other allele has an exon 55 deletion; we refer to these mice as Compound-Het mice. These mice have a phenotype resembling typical NM as has been described previously [20]. The mice were bred on a C57/Bl6J background and were housed in the animal care facility at the University of Arizona. They had unrestricted access to food and water and were maintained under a 14:10 h light/dark cycle. To administer the myostatin antibody, mRK35, we used intraperitoneal injection at a dosage of 10 mg/kg body weight. Equivalent volumes of Dulbecco’s phosphate-buffered saline served as a vehicle. The injections were performed weekly, starting at 14 days of life. At ~130 days of life, the mice were sedated using isoflurane and euthanized by cervical dislocation. Note that all mice survived until the end of the study. Muscles were isolated, weighed, and rapidly frozen in liquid nitrogen. The muscle weights were then normalized to the tibia lengths to account for body size variations. This study was conducted with the approval of the Institutional Animal Care and Use Committee (IACUC) at the University of Arizona.

### 4.2. Grip Strength

All four-limb grip-strength measurements were performed according to Tinklenberg et al. [32]. The mouse was placed on a horizontal steel mesh while the experimenter was holding its tail and the mouse voluntarily pulled away from the experimenter. Peak tensions (grams of force) from the pull were recorded on a digital force gauge (Chatillon Force Measurement DFEII, Columbus Instruments, Columbus, OH, USA). The mice were tested after weaning (~23 days of life) and at 1.5, 2, 2.5, 3, 3.5, and 4 months of life. Body weights were collected at the end of testing. Grip strength data in Figure 6C were fitted to a Y = B_max_ × X^h^/(C^h^ + X^h^) curve in GraphPad, where Y is grip strength, B_max_ is grip strength plateau, h is Hill slope, and C is a constant.

### 4.3. Running Wheel Cages

When the mice reached 4 months of age, they were housed individually in cages equipped with running wheels (Lafayette Instrument, Lafayette, IN, USA) for a period of 7 days. The running activity of the mice was monitored and recorded using Activity Wheel Monitor (v11.16, Lafayette Instruments, Lafayette, IN, USA). To analyze the data, a custom-written script in MATLAB 2023a was employed. As part of the analysis, the data from the initial two days of the recording were excluded. This decision was made because it was recognized that the mice needed time to acclimate to the running wheel cages, and their running behavior during this period might not be representative.

### 4.4. Intact Muscle Mechanics

Previous studies [20,24,48] have provided detailed descriptions of the intact muscle mechanic experiments. In brief, extensor digitorum longus (EDL) muscles were rapidly excised, and silk suture loops (USP 4-0) were tied to each tendon. The muscle was then affixed to a stationary hook, and a servomotor-force transducer, connected to an Aurora Scientific 1200A muscle system (Aurora Scientific, Aurora, ON, Canada). The muscles were submerged in an oxygenated Krebs-Ringer bicarbonate solution maintained at a temperature of 30 °C. The composition of the solution was as follows (in mM): 137 NaCl, 5 KCl, 1 NaH_2_PO_4_·H_2_O, 24 NaHCO_3_, 2 CaCl_2_·2H_2_O, 1 MgSO_4_·7H_2_O, and 11 glucose, with a pH of 7.4. To determine the optimal length (L0) of the muscle, tetanus was initially induced to eliminate any slack in the sutures. Following a recovery period, twitch responses were measured while incrementally increasing the muscle length between two twitch contractions during the passive state until the twitch forces reached a plateau.

The force-frequency relationship was assessed by subjecting the muscles to increasing stimulation frequencies (in Hz) of 1, 10, 20, 40, 60, 80, 100, 150, and 200. Between subsequent stimulations, the muscles were given recovery periods of 30, 30, 60, 90, 120, 120, 120, and 120 s, respectively. The force obtained from the experiments was converted to millinewtons (mN) and then normalized to the physiological cross-sectional area (PCSA) using the following equation: PCSA = mass (mg)/[muscle density (mg/mm^3^) × * fiber length (mm)]. For mouse EDL muscle, the physiological density of the muscle was determined to be 1.056 mg/mm^3^. The fiber length was obtained by utilizing a fiber length to muscle length ratio of 0.51 and a pennation angle of 8.3 degrees, as reported in previous studies [52].

### 4.5. Fiber Cross-Sectional Area Analysis

The analysis of muscle fiber cross-sectional area (CSA) followed the procedure described in Lindqvist et al. [20]. To summarize, extensor digitorum longus (EDL) muscles were gently stretched beyond their slack length and then pinned onto the cork. The muscles were covered with OCT (Tissue-Tek, Sakura Finetek Europe B.V., Alphen aan den Rijn, The Netherlands) and rapidly frozen in isopentane cooled by liquid nitrogen. Mid-belly sections of the muscles were obtained and embedded in OCT blocks. Ten-micrometer-thick sections were cut and placed on glass slides using a Microm HM550 cryostat (Thermo Fisher, Kalamazoo, MI, USA). The slides were stored at −20 °C for a maximum of two weeks. Prior to analysis, the slides were allowed to equilibrate at room temperature for 10 min. Each section was delineated using a hydrophobic barrier (Vector Laboratories, Burlingame, CA, USA). Subsequently, the sections were treated with 0.2% triton X-100 in phosphate-buffered saline (PBS) for 20 min on a light board. No fixation was carried out on the sections to avoid the fixative disrupting the native antigens that the antibodies recognize. Afterward, a 1-h incubation with blocking solution (2% BSA, 1% normal donkey serum) in PBS was performed in a humidity chamber at 4 °C. Primary antibodies, including laminin (1:400 rabbit L9393, Sigma-Aldrich, St. Louis, MO, USA), MHCI (1:75 IgG2b BA-F8, DSHB, Iowa City, IA, USA), MHCIIA (1:500 IgG1 SC-71, DSHB), MHCIIX exclusion ( 1:100 IgG1 BF-35, DSHB), and MHCIIB (1:50 IgM BF-F3, DSHB), were then applied to the sections and left for overnight incubation at 4 °C. Following the primary antibody incubation, the sections were washed twice with cold PBS for 30 min. Subsequently, matching secondary antibodies were applied to the sections for 3–4 h at room temperature: polyclonal Alexa Fluor 488-conjugated goat anti-rabbit [1:500 IgG (H + L) A11008, Thermo Fisher, Eugene, OR, USA], polyclonal Alexa Fluor 350-conjugated goat anti-mouse [1:500 IgG2b A211440, Thermo Fisher], polyclonal Alexa Fluor 350-conjugated goat anti-mouse [1:500 IgG1 A21120, Thermo Fisher], and polyclonal Alexa Fluor 594-conjugated goat anti-mouse [1:500 IgM (Heavy Chain) A21044, Thermo Fisher]. Post-secondary antibody washes consisted of two 30-min washes with PBS followed by two quick rinses with double-distilled water. Images of the sections were captured using an Axiocam 705 color camera on an Imager M1-microscope (Carl Zeiss, Thornwood, NY, USA) with a 5x objective [20]. The CSA analysis was conducted using the semi-automatic muscle analysis with the segmentation of histology (SMASH ver. 5) in MATLAB (R2023a) [53]. The minimum Feret (minFeret) distance, which is relatively insensitive to cutting angles during cryosectioning, was measured as it provides an indication of fiber size.

### 4.6. Western Blotting

Muscle samples were prepared following a well-documented protocol [54]. Tissues were pulverized to powder via glass Dounce homogenizers prechilled in liquid nitrogen. Tissue powder was allowed to equilibrate at −20 °C for 20 min before a 50% glycerol/H_2_O solution with protease inhibitors (in mM: 0.04 E64, 0.16 leupeptin, and 0.5 PMSF) and urea buffer (in M: 8 urea, 2 thiourea, 0.050 tris–HCl, 0.075 dithiothreitol, 3% SDS weight/volume and 0.03% bromophenol blue, pH of 6.8) were added in a 1:40:40, sample (mg):glycerol (μL):urea (μL) ratio. The solution was mixed and incubated at 60 °C for 10 min before being aliquoted and flash-frozen in liquid nitrogen. For Western blotting, solubilized samples were run on a 10% polyacrylamide gel and transferred onto polyvinylidene difluoride membranes using a semidry transfer unit (Trans-Blot Cell, Bio-Rad, Hercules, CA, USA). Blots were stained with Ponceau S to visualize the total protein transferred. Blocking, detection with infrared fluorophore-conjugated secondary antibodies, and scanning followed recommendations for the Odyssey Infrared Imaging System (LI-COR Biosciences, Lincoln, NE, USA). The following primary antibodies were used for Western blotting: anti-myostatin (1:1000, ab98337, Abcam, Cambridge, UK) and β-tubulin (1:1000, 86298S, Cell Signaling, Danvers, MA, USA). Myostatin expression was normalized to β-tubulin.

### 4.7. Myosin Heavy Chain Gel Electrophoresis

The proportion of myosin heavy chain isoforms was determined as previously described [20]. In short, myosin heavy chain isoforms were separated by sodium dodecyl sulfate polyacrylamide gel electrophoresis of muscle lysates. The stacking gel contained a 4% acrylamide concentration (pH 6.8), and the separating gel contained 8% acrylamide (pH 8.7) with 30% glycerol (*v*/*v*). The gels were run for 24 h at 15 °C and a constant voltage of 275 V. Gels were stained with Coomassie blue overnight, scanned, and analyzed with ImageJ (v1.49, NIH, USA, Bethesda, MD, USA).

### 4.8. Statistical Testing

The data is presented as means ± standard error of the mean (SEM). Statistical analysis and graph generation were performed using GraphPad Prism (version 9.5.1) software from GraphPad Software (San Diego, CA, USA). Outliers were detected using Rout’s outlier test with a 1% cutoff. For statistical testing, we utilized two-way ANOVA with multiple-comparison correction to assess treatment effects. Post hoc tests, specifically Tukey’s or Šídák’s tests as recommended by GraphPad, were employed for pairwise comparisons. To determine the significance of differences between curves, Gompertz’s function was fitted to body weight curves using least square regression, and the extra sum-of-squares F-test was utilized.

## Figures and Tables

**Figure 1 ijms-24-15124-f001:**
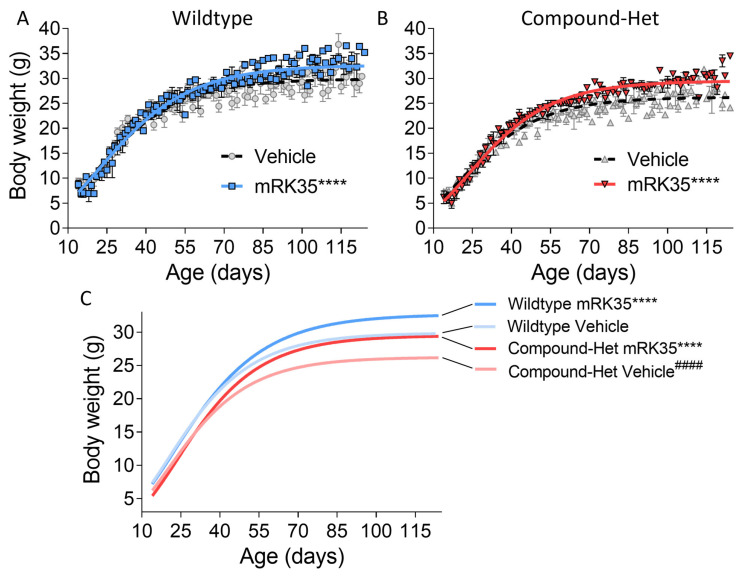
mRK35 treatment increases body weights in both wild-type (**A**) and Compound-Het (**B**) mice. (**C**) Growth curves for all groups overlayed. mRK35 increased the body weights of the Compound-Het mice to that of untreated wild-type mice. A Gompertz’s growth curve was used to fit body weights. **** *p* < 0.0001 indicates significantly different curves vs. vehicle-treated mice of the same genotype. #### *p* < 0.0001 indicates a significantly different curve vs. vehicle-treated wild-type mice. N-values: WT Vehicle: 14; WT mRK35: 10; Compound-Het Vehicle: 11; Compound-Het mRK: 8.

**Figure 2 ijms-24-15124-f002:**
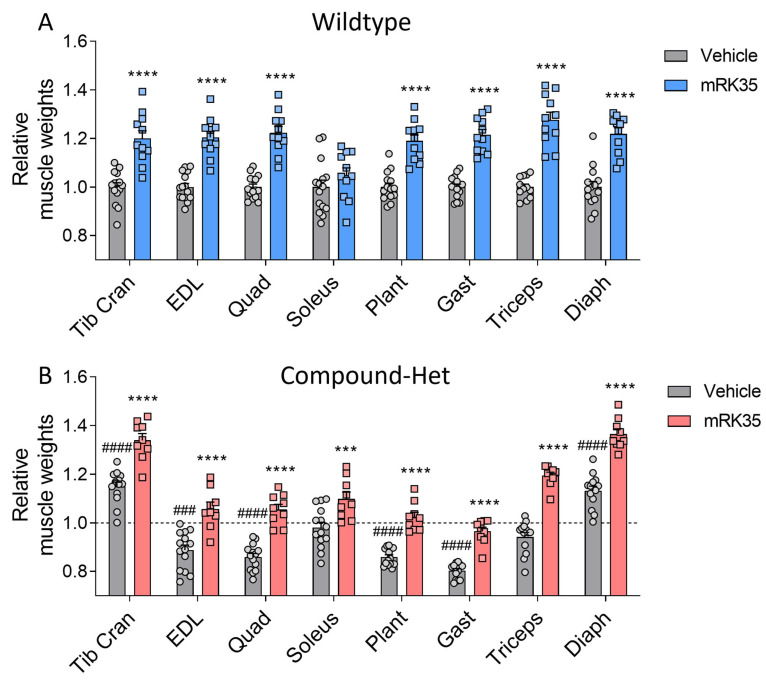
mRK35 treatment resulted in hypertrophy of multiple skeletal muscles in both wild-type (**A**) and Compound-Het (**B**) mice. Graphs show the relative change in muscle weights normalized to tibia lengths. The dashed line in B indicates vehicle-treated wild-type mice. Two-way ANOVA with Tukey’s post hoc test was used for statistical testing. *** *p* < 0.001 and **** *p* < 0.0001 vs. vehicle-treated mice of the same genotype. ### *p* < 0.001 and #### *p* < 0.0001 vs. vehicle-treated wild-type.

**Figure 3 ijms-24-15124-f003:**
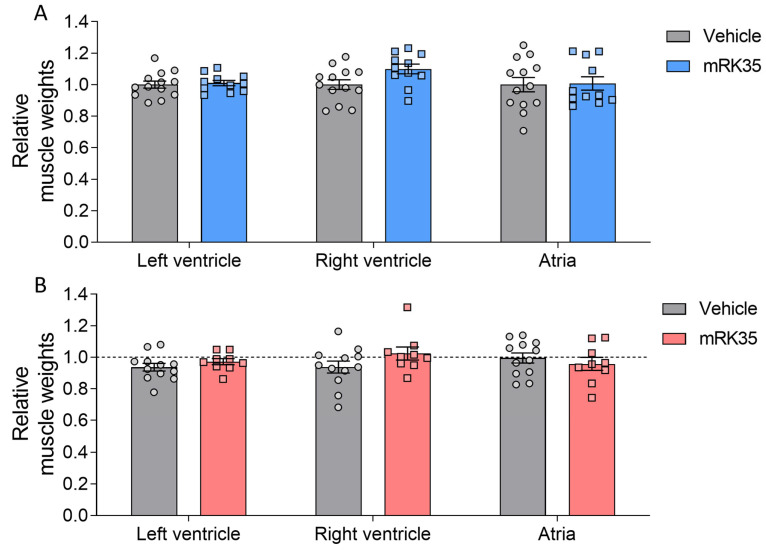
No effect of mRK35 treatment on cardiac muscle tissue weights in wild-type (**A**) and Compound-Het (**B**) mice. Graphs show the relative change in muscle weights normalized to tibia lengths. The dashed line in B indicates vehicle-treated wild-type mice. Two-way ANOVA with Tukey’s post hoc test was used for statistical testing.

**Figure 4 ijms-24-15124-f004:**
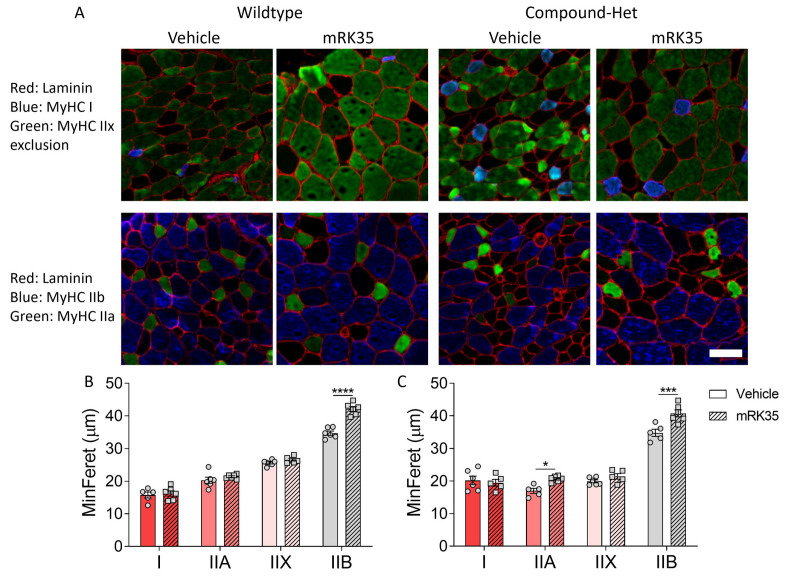
mR35 treatment increases primarily the minFeret diameter of type Iib fibers. (**A**) Immunofluorescence images of the different fiber types found in EDL muscles. Top row; red: laminin; blue: type I; green: type IIx exclusion (non-stained fibers are type IIx). Bottom row; red: laminin; green: type IIa; blue: type IIb. (**B**) MinFeret diameter in wild-types. (**C**) MinFeret diameter in Compound-Hets. Two-way ANOVA with Šídák’s post hoc test was used for statistical testing. * *p* < 0.05, *** *p* < 0.001 and **** *p* < 0.0001 vs. vehicle-treated animals of same genotype. Scale bar: 50 µm.

**Figure 5 ijms-24-15124-f005:**
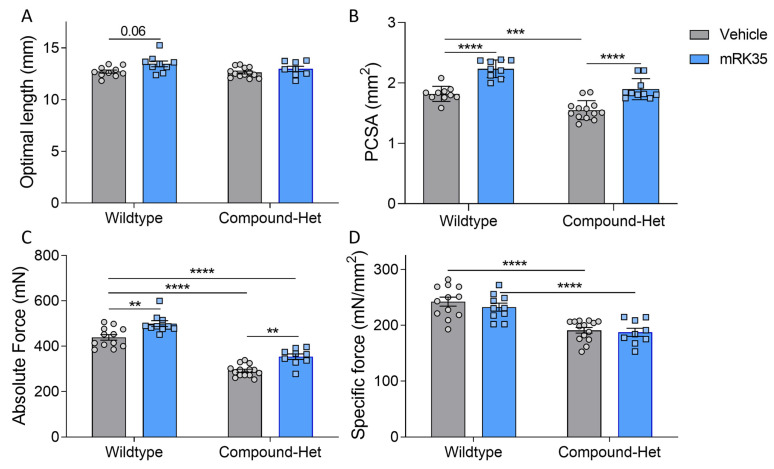
mRK35 treatment enlarges the physiological cross-sectional area of EDL muscles and increases force production. (**A**) Optimal length for force production. *p* < 0.01 for treatment effect in a two-way ANOVA. (**B**) Physiological cross-sectional area (PCSA) at optimal length. (**C**) Absolute force. (**D**) Specific force (absolute force normalized to cross-sectional area). Two-way ANOVA with Tukey’s post hoc test was used for statistical testing. ** *p* < 0.01, *** *p* < 0.001 and **** *p* < 0.0001.

**Figure 6 ijms-24-15124-f006:**
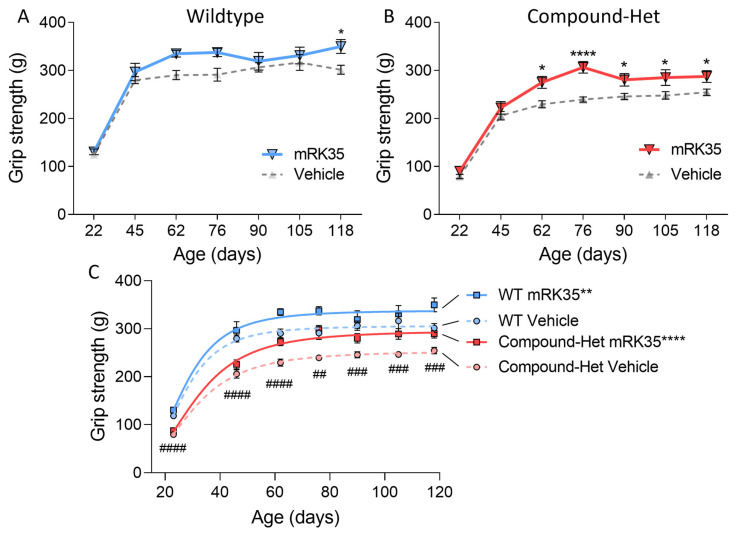
Increased grip strength in mRK35-treated mice. (**A**) In wild-type mice, mRK35 only increased grip strength at 4 months of life. (**B**) From two months of life and onwards Compound-Het mice treated with mRK35 displayed greater grip strength. (**C**) Grip strength for all groups at all time points. Two-way ANOVA with Šídák’s post hoc test was used for statistical testing. In C, grip strength at the plateau of the fitted curves is compared (see methods for details). * *p* < 0.05, ** *p* < 0.01, and **** *p* < 0.0001 vs. vehicle-treated mice of the same genotype. ## *p* < 0.01, ### *p* < 0.001, and #### *p* < 0.0001 for vehicle-treated Compound-Het vs. vehicle-treated wild-type.

## Data Availability

All data for this study is available upon reasonable request.

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
