# Peer review of "Pharmacological Inhibition of Myostatin in a Mouse Model of Typical Nemaline Myopathy Increases Muscle Size and Force"

_ijms, 2023, doi:10.3390/ijms242015124_

Round 1

Reviewer 1 Report

Comments and Suggestions for Authors

In this study, the authors tried to establish the link between the progressive inhibition of muscle growth following aging in the nemaline myopathy and contradictory induced acceleration of muscle growth by an inhibition of myostatin. Myostatin is a negative regulator of excessive muscle hypertrophy in the normal physiological condition, thus the idea of a contradictory comparison is interesting.

However, it seems that this work was done as the continuation of the story of “ref. 25” by the same first and last authors using the same mouse model. Thus, a basic change of the notion is simply the addition of weekly administration of mRK35 (a myostatin antibody) to block myostatin. Although several major analyses, which have been carried out in previous work (25) such as molecular analysis and electron microscopy, have not been done in this work. Additionally, for the functional analysis, the previous study used in vivo (in-situ) contractile analysis under electrical stimulation, but the present study used in vitro condition. 

Therefore, this work almost composed the data of numerical changes in body and muscle mass and functional (in vitro contractions and running wheel and grip strength). Thus, the present work gives the impression of being precarious and unsatisfactory as in the molecular sciences. Although it can be supposed from the previous work (ref 25) that the authors have various techniques of molecular science.

Therefore, my questions to be clarified are as follows;

1)      Why the authors did not measure an expression of myostatin in the muscles.

2)      Why only used the EDL muscle? This is minor muscle in the mice hindlimb. I want to know how are the other major muscles (mass and histological changes).

3)      Similarly, why only used in vitro contractile capacity of EDL? In vivo (in-situ) functions as the plantar flexors via stimulation of sciatic nerve may be much proper for physiological comparison.

4)      When I read the method of both previous and this work, I was surprised and unbelievable that the authors performed immunohistochemistry for the skeletal muscle without fixation. If this is the case, all immunohistochemical data should be considered invalid, because of the protein diffusion. In the EM method of previous work, the authors correctly described tissue fixation, but not for the histochemistry. The authors used fresh frozen cryosections both now and previously, so tissue fixation is crucial. Therefore, the histology data of this group would not be acceptable in a fairer manner.

5)      The animal model in this study used a mouse model with compound heterozygous nebulin mutations. Therefore, Gomori’s modified Trichrome Staining, which is basic/typical staining of NM, after proper fixation should be necessary, and this is very suitable for determining fiber diameters. Why not at this time, although a fixation has not been clear at last time.

6)      Furthermore, why is electron microscopy not currently available?

7)      Including the above three issues, the effects of myostatin inhibition on the basic symptoms of nemaline myopathy did not clear up at all. Therefore, only the body and muscle mass, the in vitro contractile capacity of EDL and the physical functioning of the wheels and grip strength are available at this time. There is likely an unacceptable level down from 25.

8)      How was the lifespan time for all groups?

9)      In the third paragraph of the discussion, I recommend using the cross-section of the entire lower hind limb, including all the muscles that are composed. This could resolve the matter relating to fiber types.

10)   I completely agree with the last sentence of the third paragraph of the discussion; therefore, I am expecting additional data because the authors have the potential. Additional data should be provided, even a little more.

11)   Based on the results of this study, the fourth paragraph of the discussion appears to be excessively detailed currently.

12)   In the fifth, compared to ref 49, the component of current data is too small.

13)   I think that description of the statistical significance may be sufficient only using P<0.05. Descriptions of further values has no meaning and disturb the clearance of each graph.

Generally, cryosection of fresh frozen tissue should be fixed 2-4% paraformaldehyde before staining. However, without this explanation of the method is a critical mistake.

Reviewer 2 Report

Comments and Suggestions for Authors

Summary

This study investigated the effect of myostatin inhibition by a myostatin antibody, mRK35, in a mouse model of nebulin-mutated nemaline myopathy (Compound-Het). Intraperitoneal injections of mRK35 successfully increased in muscle mass and force in both wild-type and Compound-Het mice without apparent cardiac changes. The results provide evidence that myostatin inhibition can be a candidate therapy for nemaline myopathy. The manuscript is well documented to describe the methods and results that support the authors’ claim. I pointed out on issue below to make this study perfect for publication in the International Journal of Molecular Sciences.

Comments

1.          Figure 4: Scale bars should be displayed.

2.          Figure 4: The composition (% ratio) of the myofiber types I, IIA, IIX, and IIB appears to be altered by mRK35. It is generally known that myostatin is involved in myofiber type composition (e.g. Girgenrath, Muscle Nerve, 2005; 31: 34. Hennebry, Am J Physiol Cell Physiol, 2009; 296: C525). Furthermore, myofiber type transformation has been observed in nemaline myopathy (e.g. Miike, Brain Dev, 1986; 8: 526. Gurgel-Giannetti, J Child Neurol, 2003; 18: 235). The myofiber type composition of the mice used in this manuscript should be quantified and statistically analyzed. And the relationship between phenotypic change and myofiber type transformation by myostatin inhibition needs to be discussed.

Minor points

3.          Line 10: “a third” would be “one-third” as in the line 35.

4.          Lines 57-79: The font color should be black.

5.          Lines 93 and 138: “similar” should be “same” as in the lines 116 and 191.

6.          Line 192: Statistical characters (# and *) should be fully explained (vs ???).

7.          Line 306: “ser6366ile” should be “Ser6366Ile”. Or, “S6366I” would be better because other amino acid substitutions are described in one-character as in the lines 253-256.

8.          Line 394: “SEM” should be defined.

Round 2

Reviewer 1 Report

Comments and Suggestions for Authors

For response 4 and 14; fixation should be done even after treatment of the first antibody.

For response 13; I still cannot feel a biological meaning.

Reviewer 2 Report

Comments and Suggestions for Authors

The authors revised the manuscript according to the reviewer's comments.